# Degradation of Styrenic Plastics during Recycling: Accommodation of PP within ABS after WEEE Plastics Imperfect Sorting

**DOI:** 10.3390/polym13091439

**Published:** 2021-04-29

**Authors:** Charles Signoret, Pierre Girard, Agathe Le Guen, Anne-Sophie Caro-Bretelle, José-Marie Lopez-Cuesta, Patrick Ienny, Didier Perrin

**Affiliations:** 1Polymers Composites and Hybrids (PCH), IMT Mines Ales, 30100 Ales, France; charles.van.signoret@orange.fr (C.S.); pierre.girard@mines-ales.org (P.G.); agathe.le-guen@mines-ales.org (A.L.G.); jose-marie.lopez-cuesta@mines-ales.fr (J.-M.L.-C.); 2LMGC, IMT Mines Ales, Université Montpellier, CNRS, 30100 Ales, France or anne-sophie.caro@mines-ales.fr (A.-S.C.-B.); patrick.ienny@mines-ales.fr (P.I.)

**Keywords:** mechanical recycling, impact resistance, WEEE, polymer compatibilization, design of experiments, surface response methodology

## Abstract

With the development of dark polymers for industrial sorting technologies, economically profitable recycling of plastics from Waste Electrical and Electronical Equipment (WEEE) can be envisaged even in the presence of residual impurities. In ABS extracted from WEEE, PP is expected to be the more detrimental because of its important lack of compatibility. Hence, PP was incorporated to ABS at different rates (2 to 8 wt%) with a twin-screw extruder. PP was shown to exhibit a nodular morphology with an average diameter around 1–2 µm. Tensile properties were importantly diminished beyond 4 wt% but impact resistance was decreased even at 2 wt%. Both properties were strongly reduced as function of the contamination rate. Various potential compatibilizers for the ABS + 4 wt% PP system were evaluated: PPH-g-MA, PPC-g-MA, ABS-g-MA, TPE-g-MA, SEBS and PP-g-SAN. SEBS was found the most promising, leading to diminution of nodule sizes and also acting as an impact modifier. Finally, a Design Of Experiments using the Response Surface Methodology (DOE-RSM) was applied to visualize the impacts and interactions of extrusion temperature and screw speed on impact resistance of compatibilized and uncompatibilized ABS + 4 wt% PP systems. Resilience improvements were obtained for the uncompatibilized system and interactions between extrusion parameters and compatibilizers were noticed.

## 1. Introduction

The consumer society led to the disposal of 4.9 billion metric tons of polymeric waste in the environment from 1950 to 2015 according to Geyer et al. [1]. Authors anticipated, with optimistic measures taken in favor of recycling and incineration, that the corresponding cumulative waste stock should stabilize around 12 billion in 2050. Furthermore, Robinson [2] reported that E-waste (electronic waste) is estimated at around 20–25 million tons of worldwide production per year. Waste Electrical and Electronic Equipment (WEEE) is a more generic term and also includes traditionally strictly electrical equipment such as refrigerators or kettles. Plastics are reported to represent from 10 to 35 wt% of WEEE [3,4,5,6,7,8] even if it varies much between studies and WEEE types (small/large appliance for instance).

Recycling is viewed as one of the main routes to prevent plastic pollution, and, in the same way, to produce new materials [9,10]. However, WEEE plastics (WEEP) are more complex, diverse and contain generally more additives than other plastics [5,11,12,13]. Stenvall et al. [5] found that HIPS, ABS and PP were the major constituents of their studied WEEP samples, respectively, at 42, 38 and 20 wt%. Maris et al. [14] found 26, 29 and 22 wt% for these three categories of plastics. Additionally, an important fraction of WEEP is dark-colored, preventing an efficient sorting thanks to current technologies such as NIR-HSI [15,16]. Yet, sorting is mandatory to achieve interesting properties since most polymers are said incompatible, leading to immiscibility and poor mechanical properties [11,16,17]. However, several technologies are currently in development, such as MIR-HSI [18,19,20], Terahertz spectroscopy [21,22] or LIBS [23,24,25], which could then enable profitable recycling of WEEP.

Nevertheless, sorting alone can be insufficient as perfect purities are impossible to achieve, especially in an economic way [15]. However, scientific literature, about recycled materials containing a small fraction of impurities close to what an automatized sorting could produce, is scarce. A consistent review from Vilaplana and Karlsson [6] shows that most studies concern systems where the second component is present above 10 wt% or, more frequently, above 20 wt%. Thus, Perrin et al. [26] studied the influence of different possible contaminants, PS, ABS and PP, on the mechanical properties of HIPS. Impact properties were the most reduced. They concluded that ABS was tolerable upon 4 wt% and PP upon 1 wt% only.

Since ABS/HIPS blends were thoroughly studied [27,28,29] as well as contaminated HIPS [26], another interesting case would be ABS contaminated with PP. Especially, ABS presents several interesting properties compared to HIPS (higher chemical resistance, Young’s Modulus, resilience and stress at peak [27,28,29,30] and thus could present better added-value upon recycling.

ABS is basically a copolymer of styrene and acrylonitrile (SAN) where polybutadiene (PB) nodules are present to dramatically improve impact properties through the slowdown of crazes propagation, preventing fast crack formation [31,32,33,34]. Thus, differences in PB concentrations lead to very different impact properties, but also viscosities, between several commercial grades. PPC stands for polypropylene copolymer. It is differentiated from PPH (Homopolymer) as ethylene is present within its chemical formula. However, it is not always clear if it is by copolymerization [35,36] through EPR or EPDM introduction [36,37]. In either case, it results in improved impact properties and lower modulus. Sadly, the difference between PPH and PPC is rarely highlighted during WEEP characterization even though it is easily possible through FTIR [30]. In WEEE, PPC seems more adequate, however.

The addition of adequate compatibilizers, often copolymers or grafted polymers, leads to morphology reduction thanks to interfacial tension decrease (and thus capillary number increase) and/or coalescence prevention through steric stabilization as compatibilizer “shells” can form around the dispersed phase droplets [17,38,39]. In terms of ABS/PP systems, most authors worked on PP-rich systems, often around 70/30 or 80/20 [40,41,42,43,44,45,46,47]. Kum et al. [43] tried PP-g-SAN on ABS/PP systems by 10 wt% steps and found reduced nodular morphology and enhanced mechanical properties on most of the range even if the overall is still far below pure ABS. Patel et al. [46] added home-made PP-g-2-HEMA (2-hydroxyethyl methacrylate) and PP-g-AA (acrylic acid) to ABS/PP at different ratios and highlighted morphological refinement. Tensile and flexural properties, but also impact resistance, were improved in most of the range, with the notable exception of ABS with 10 wt% PP on the last property. Deng et al. [41] compared the effects of PP-g-MA and home-made PP-graft-cardanol on a PP70/ABS30 system and found that the first one was more efficient to improve mechanical properties, mainly tensile and flexural strengths, thanks to morphological refinement. Bonda et al. [40] showed that ABS impact and tensile properties were strongly diminished by 10 wt% PP and that addition of PP-g-MA, SEBS-g-MA and EAO (ethylene α-olefin) could improve impact properties of a PP80/ABS20 blend through morphological refinement to the detriment of tensile properties. They proposed a dipole–dipole interaction between a C=O group of maleic anhydride sites and an ABS nitrile group to explain the improvement in properties by the decrease in interfacial tension. Ibrahim et al. [42] later proposed a real reactivity between the two moieties where the cycle of maleic anhydride could open up to form an eight-membered cycle with acrylonitrile, thus creating in situ copolymers. Lee et al. [44] evaluated SEBS and SEBS-g-MA on a PP70/ABS30 system and found improved impact properties but decreased tensile strength. The second compatibilizer was found to be more efficient, especially after thermal ageing where it preserved properties contrary to the first one. Finally, Tostar et al. [47] worked on unsorted WEEP as described by Stenvall et al. [5] of the same research group (42 wt% HIPS, 38 wt% ABS, 10 wt% PP, the rest being PE, PUR, PC, etc.). They found SEBS much more efficient than SEBS-g-MA, EPDM-g-SAN, PP-g-MA or gamma irradiation to improve mechanical properties, especially strain at break. It also greatly improved the impact properties.

For other polymer blends, several authors highlighted the importance of mixing parameters on phase dispersion. Plochocki et al. [48] studied an LDPE/PS (2/1) blend realized with different industrial processes and found that PS domain size versus mixing energy gave a hyperbolic curve. They reported that dispersion followed a first order rule with shear rate, but coalescence a second order one as shearing also increase droplets collision probability and thus coalescence. Sundaraj and Macosko [39] studied different polymer blends realized in a batch mixer and also found a hyperbolic curve for dispersed phase diameters versus shear rate in the case of uncompatibilized systems. They explained this phenomenon rather by different rheological responses of both phases toward shear rate. They also highlighted that copolymers, especially formed in situ, enables interfacial tension decrease and coalescence prevention. Therefore, mixing parameters can have an important impact on blends morphologies, and thus properties, especially toward impact [49].

It appears from the literature that, in a WEEP recycling context, ABS/HIPS and contaminated HIPS have been already studied. Additionally, many authors studied PP-rich blends to find that most properties of pure ABS were lost by blending, especially impact properties. Compatibilization effectiveness was often explained through morphological refinement but was mainly operant in PP-rich systems. Thus, in a first part, the present study aims to evaluate the impact of PPC within ABS with incorporation rates from 2 to 8 wt% through twin-screw extrusion. The second part relates to compatibilization trials on an ABS/PP (96/4) system, with the purpose to retrieve properties close to virgin ABS. Morphological assessment, through SEM, and impact properties, through instrumented Charpy impact, are at the core of considerations. Tensile properties were also evaluated. In the literature, PP-g-MA (here differentiated between PPH-g-MA and PPC-g-MA), SEBS and PP-g-SAN were found to have positive effects and are applied here to an ABS-rich system. TPE-g-MA and ABS-g-MA were also evaluated. Finally, the third part is about a Design Of Experiments Response Surface Methodology (DOE-RSM) to model impact resistance of ABS/PP (96/4) as a function of speed and temperature extrusion, with or without selected compatibilizers. Ultimately, this was used to localize maximal resistances.

## 2. Materials and Methods

### 2.1. Materials

ABS used for this work was Terluran GP22 provided by Styrolution and PPC was PHC27 from Sabic. Five compatibilizers were kindly provided by Velox: PPH-g-MA (Bondyram 1001), PPC-g-MA (Bondyram 1001 CN), ABS-g-MA (Bondyram 6000) and TPE-g-MA (Bondyram 7108) and SEBS (Tuftec P2000). The last one was PP-g-SAN (Modiper A3400) kindly supplied by Modiper. Figure 1 shows their chemical structures.

Maleic anhydride (MA) grafting rates were about 1 wt% for every grafted polymer used here. As reported in the introduction, MA-grafted polymers were chosen since this moiety could promote compatibility towards acrylonitrile. More precisely, PP-g-MA grades were chosen because of results found in the literature and as it is a very common compatibilizer for many systems (Maris et al., 2018). PPH and PPC were differentiated in case. PPC, which is less common, could show better compatibility with the minority phase. The TPE used here was an olefinic thermoplastic elastomer with a strong ethylene base. It could show compatibility or even miscibility with polyolefins. It could also act as an impact modifier. Finally, a soft interphase between PP nodules and the matrix could be created. SEBS, here with a weight S/EB ratio of 67/33, can also fulfil these roles, even to a larger extent. This specific ratio was chosen as ABS is the major component. Available SEBS-g-MA grades were all EB rich, so these compatibilizers were not evaluated. Finally, PP-g-SAN appears theoretically to be the optimal polymer for such a system as it could be placed at the interface and bring adhesion.

### 2.2. Polymer Processing

ABS was dried at 80 °C for at least 16 h before processing. Pellets were mixed by hand and were then processed with a 1200 mm Clextral twin-screw extruder of type BC21 (L/d = 48) at 250 rpm and 220 °C along the screw, unless otherwise specified. The nozzle was 5 mm and feed speed was at 6 kg/h thanks to a K-Tron KGx-2 weighing dozer from Coperion. Extrudate was then pelletized for further transformations.

Pellets were then injected into dogbone specimens corresponding to ISO 3167 thanks to a Krauss-Maffei 180/CX 50 molding injection press at 230 °C for plasticization and mold maintained at 40 °C. Pressure was maintained at 450 bars for 12 s following injection, then 400 bars for 12 s again before ejection. Specimens for Charpy impact, 80 × 10 × 4 mm^3^, were cut off from dogbones.

Amounts of PP are given in wt% as this simulates what recyclers receive and should deal with. Amounts of compatibilizers are given in phr as this is what they should add to the material to give it interesting properties.

As processing itself can affect the material, each study included at least one uncontaminated ABS batch that went through the same transformations as the others. This batch is identified as “Vext” or “0 wt%” to differentiate it from a really virgin batch, “V” which consists of directly injected pellets.

### 2.3. Mechanical Characterization

Results from both Charpy impact and tensile tests were processed thanks to Matlab 2018a from MathWorks and then plotted with OriginPro 9.1 from OriginLab Corp, Northampton, MA, USA. Boxplots plotted in these works represent quartiles for box limits, medians as a horizontal line, means as crosses and minima and maxima as whiskers. Values are also plotted as dots at the left of each corresponding box.

### 2.4. Charpy Impact

Instrumented impact tests were carried out thanks to an impact drop tester Ceast 9340 from Instrom. Both notched (1eA) and unnotched (1eU) Charpy impact tests were performed according to ISO 179. The two tests give different but complementary information about the material. Whereas unnotched tests cumulate crack initiation and propagation, notched tests focus on the second part. Additionally, whereas unnotched tests are very sensitive to skin effects, notched tests really evaluate the core of the material. *Appendix A* shows force–displacement curves of notched specimens at two different velocities which are chosen by changing drop height. 2.9 m/s, the velocity recommended by ISO 179, generates relatively important vibrations compared to 0.8 m/s because impact energy is too important. The break energy is calculated by integration. However, force does not really return to 0 N. Therefore, 15% of maximal force was chosen in the present work as a cut-off parameter. However, vibrations which can be irregular between batches can falsely shorten integration at 2.9 m/s; 0.8 m/s was thus chosen for all notched Charpy impact tests in this study.

### 2.5. Tensile Tests

Tensile tests were performed according to the ISO 527 standard and thanks to a Z010 Zwick press equipped with a 2.5 kN load cell and a Clip-on extensometer. All tensile properties were measured at a 10 mm/min crosshead speed. Young’s moduli were determined between 0.05% and 0.25% of deformation. Ultimate strains were measured according to the crosshead displacement.

### 2.6. Characterization of Morphologies

Observations of blend morphologies were performed thanks to a scanning electron microscope (Quanta FEG 200, Thermo-Fisher Scientific, Waltham, MA, USA) on cryo-fractured dogbones samples (orthogonally or 45°) and on post-mortem Charpy impact fragments after carbon metallization at magnifications of ×5000 and ×25,000. Only secondary electrons were considered as chemical differences are too weak between considered polymeric materials.

### 2.7. Design of Experiments

Design of Experiments (DOE) builds and visualizations were carried out using the Nemrodw^®^ 2015 software. Two responses (Y) were considered: mean break energies from notched Charpy impact, and from unnotched Charpy impact. The DOE was applied to an uncompatibilized ABS + 4 wt%PP system, or with SEBS (1, 2 or 3 phr), PP-g-MA (0.5, 1.0 or 1.5 phr) or PP-g-SAN (0.5, 1.0 or 1.5 phr). Twenty experiments were performed, each resulting in 15 Charpy’s impact tests performed on notched and unnotched specimens. A Response Surface Methodology (RSM) was applied to evaluate the influence of several process parameters, extrusion temperature, screw speed and compatibilizer addition rate, on the two responses. The applied interaction mathematical linear model is given in Equation (1):(1)$$Y=b_{0}+b_{1}\cdot X_{1}+b_{2}\cdot X_{2}+b_{3}\cdot X_{3}+b_{1-2}\cdot \left(X_{1}\cdot X_{2}\right)+b_{1-3}\cdot \left(X_{1}\cdot X_{3}\right)+b_{2-3}\cdot \left(X_{2}\cdot X_{3}\right)$$
where X_1_ is the extrusion temperature (200–240 °C), X_2_ the screw speed (200–300 rpm) and X_3_ the additive loading rate (phr, depends on considered additive) and different b_n_ constants (Equation (1): Interaction mathematical linear model equation).

This model represents influence of different parameters at first order and 1st order interactions.

## 3. Results and Discussion

### 3.1. Impact of PP on ABS for Various Incorporation Rates

To evaluate the tolerance of PPC impurities from ABS, PPC was added to ABS through extrusion at increasing rates: 2, 4, 6 and 8 wt%.

#### 3.1.1. Impact Properties

For both notched and unnotched impact tests, a first slight diminution is due to processing alone (from black to blue on Figure 2). Then, impact resistance progressively decreases with PPC incorporation rate, from about 16.5 to below 10.0 kJ/m^2^ for notched impact (−40%), and about 80 ± 20 kJ/m^2^ to about 15 kJ/m^2^ for unnotched impact (−80%). Interestingly enough, 2 wt% does not affect unnotched impact resistance much as it really begins to decrease at 4 wt%, which also corresponds to a more important drop in notched impact resistance.

On notched force–displacement curves of Figure 2, the whole bell-shaped curve gradually collapses, with the maximal force decreasing from about 230 N to about 175 N. Especially on the 8 wt% PP batch, the bell shape gets more asymmetrical, with a more vertical force drop after the maximum. It hints at a more brittle behavior. On unnotched impact curves, the behavior is quite different. The force first increases to reach a plateau in load. On virgin batches, the drop of force is dispersed in displacement values because of a more statistical occurrence of fracture. The whole plateau in load is linked to irreversible behavior, the material undergoing damaged plastic behavior and developing a multitude of crazes, resulting in visible whitening shown further below. As the break energy is calculated by the integration of the shown curves, this break displacement dispersion results in break energy dispersions as seen in the boxplots. Progressively with PP contamination, this plateau shortens and so failure occurs in a narrowed displacement range, especially from 6 wt%. Thus, break energies become smaller and closer. Finally, at 8 wt% of contamination, force does not even reach the plateau level and curves are perfectly superimposed, adopting a brittle behavior similar to what can be obtained with SAN (*Appendix A*) or PMMA [50]. Especially, 6 wt% contaminated ABS is even less resistant than SAN Luran 368 R. Finally, for both notched and unnotched impact, the initial force slope gets slightly weaker with contamination. As the Charpy impact is basically a high-speed flexion test, a flexion modulus can be extracted from these curves as performed by other authors [30] on photodegraded and recycled HIPS. However, in the present case, comparisons to Young’s moduli measured from tensile test were not conclusive even if general trends are more or less in agreement (*Appendix A*).

Figure 3 shows broken specimens after unnotched Charpy impact for the same batches. Hardly visible on the picture, the general hue of specimen gets slightly whiter with PP contamination.

As seen on bottom specimens, the whitening behavior during impact develops at the opposite of the impactor, in the tensile part of the bending test specimen (from the underside to the neutral line), displaying a rather triangular whitened shape. The fracture is rather planar and directly orthogonal to the direction of uniaxial stresses. With PP, the whitening progressively fades, almost invisible at 4 wt%, and is totally nonexistent beyond this, as corroborated by the absence of plateau a (Figure 2). The fracture surface is also getting more and more uneven and more often oblique. Additionally, on several fragments, little spikes protruding from the center of break surfaces can be seen as framed in Figure 3. Beginning at 6wt%, this heterogeneity is even more important at 8 wt%, represented by rods up to 2 mm thick as shown and discussed with SEM characterization in 0.

Whitening is commonly explained in impact modified styrenics by light diffusion due to important craze netting originating from the boundary region of PB to the matrix (Yokouchi et al., 1983). The absence of whitening proves that energy dissipation from PB is inhibited. The rougher aspect of break surface underlines the heterogeneity of the sample discussed above.

#### 3.1.2. Tensile Properties

Tensile properties are presented in Figure 4. The magnification on stress peak shows its progressive collapse beginning clearly from 4 wt%. As with unnotched Charpy impact, strain at break is very dispersed because of plastic damage behavior.

Figure 5 gives values extracted from these curves. Overall, the Young modulus is slightly impacted but its decrease is more pronounced at 8 wt%. Maximal stress diminished almost linearly with PP concentration, except at 8 wt%. Whereas strain at break is rather dispersed and does not display clear evolution, stress at break steadily decreases, especially above 2 wt% and becomes more and more dispersed. At 6 and 8 wt% however, the strain at the break is significantly smaller, as is clearly visible on the full curves as well (Figure 4). This can be linked to the heterogeneities seen in Figure 3 beginning at these concentrations.

#### 3.1.3. Morphology

At the studied concentrations, PP adopts a nodular morphology (Figure 6). Cryofractures at 45° (*Appendix A*) confirms the spherical shape as pictures are very similar from both break angles. At 2 wt%, nodules are very scarce and their diameters are comprised between 0.5 and 1.5 µm. At 4 wt%, nodules are more notably more numerous and general diameters go up to 0.5–2.0 µm and some especially big nodules are measured between 3.0 and 3.5 µm. At 6 wt%, the nodules concentration continues to increase but measured diameters are in the same ranges.

As seen in Figure 3, macroscopic heterogeneity is visible to the naked eye beginning at 6 wt% PP. The phenomenon is even more important at 8 wt% as shown in Figure 7. A central part (pink arrow) was protruding from the fracture surfaces. On some specimens, it is a simple spike as in Figure 3 and on others, it is an ellipsoidal cylinder.

Figure 8 shows SEM pictures on cryofractured samples. The right bottom corner SEM picture represents the sample at a magnification low enough for its full thickness (4 mm) to be visible. Fractures can be seen near the surfaces, magnified on the right-bottom corner, separating “core” from the “skin”. The “core” displays different fracture behaviors and the part indicated in pink seems to result from a more fragile break than the areas around it. Stronger magnifications indicate that nodules could be coarser and more numerous in the central part (top-left picture) than in the rest of the sample (top-right picture). Due to the rough fracture surface due to cryofracture, image analysis could be not performed to corroborate this.

Fellahi et al. [51] described a complex system in HDPE/PA6 blends, as skin–subskin–core. However, the “subskin” of their system was of the same order of magnitude as the “skin”. Other authors [52,53,54,55,56,57] also described “skin-core” as rather progressive, with intermediate layers, issued from important differences in cooling and shearing during injection-molding.

It was found through higher magnifications that the skin is devoid of any nodules. On their impact-modified PP, Karger-Kocsis and Csikai [52] found a “Surface Matrix Layer” but differentiated it from the skin which they defined asfIGURE6 displaying fibrillar morphology. The “skin” can be measured in Figure 8 at roughly 200 µm thick on the SEM picture, coherently with micrometer measurements performed on the few broken specimens from both tensile and impact tests where the “skin” was protruding from the fragments. FT-IR analysis (ATR, 4 cm^−1^ resolution, 16 scans, Vertex 70, Bruker) confirmed that skin was ABS (spectra presented in *Appendix A* and Appendix B). Other contaminated specimens also display “skins” of pure ABS, roughly from 100 to 200 µm thick, from both SEM and micrometer measures. Especially, they keep fragments attached after notched Charpy impact at low speed in the form of a flexible hinge (noted “H” in ISO 179 standard).

#### 3.1.4. Conclusions on PP Contamination of ABS

It is confirmed through these first works that ABS and PP are incompatible as PP forms a nodular dispersion, which can result in heterogeneities at the millimeter scale for highest contamination rates. Tensile properties are mostly preserved below 4 wt% contamination, with just a small maximal stress decrease. Beyond this, elongation at break is also strongly diminished. Impact resistance is reduced in a stronger measure, by about 40% for notched impact, and 80% for unnotched impact. This drop also begins at lower concentrations, at 2 wt% for notched and at 4 wt% for unnotched. From 6 wt%, the force–displacement curves of unnotched impact demonstrate a brittle behavior. Industrial sorting nominal purity is recommended at 5 wt% max as reported by Beigbeder et al. [15], which is strongly justified by present results. However, 4 wt% contaminated samples correspond already to a drop in impact properties and the beginning of morphology coarsening. Thus, this concentration was chosen for the following compatibilization trials.

### 3.2. Compatibilizers Preselection

Six potential compatibilizers, PPH-g-MA, PPC-g-MA, ABS-g-MA, TPE-g-MA, SEBS and PP-g-SAN, were tested on ABS contaminated with 4 wt% of PP by additions at 1, 2 and 3 phr to check the effect of additive concentration.

#### 3.2.1. Impact Properties

Figure 9 represents the notched Charpy impact on these samples. The first two boxplots correspond to references, virgin but extruded ABS and ABS with a contamination of 4 wt% PP. All compatibilizers led roughly to intermediate properties between these two references at most concentrations, except for PP-g-SAN. This last compatibilizer seems to entail a degradative behavior of PP at 2 and 3 phr. All maleic anhydride compatibilizers present effects similar to each other, around 14.5 kJ/m^2^. A negative effect of concentration is perceived for all of the g-MA compatibilizers, especially PP-g-MA, homopolymer or copolymer. Even worse, some samples are far below the others in terms of energy. Force–displacement curves (*Appendix A*) inspection proves that it is due to a premature break, in a more brittle manner. All these concentration-related negative effects can be explained as the compatibilizer is added in excess and creates another non-dissipating phase, playing the role of a contaminant.

ABS-g-MA has thus a limited negative effect, as it should not be incompatible or even immiscible with the matrix. However, it is surprising that even TPE-g-MA has this negative effect as it is an elastomer and could simultaneously play an impact modifier role. It could be that its impact properties and/or its compatibility toward the matrix are insufficient, leading to the formation of a separated phase. Finally, SEBS displays the best results, just slightly above the others. Impact reinforcement increases with SEBS concentration, improbably because of its compatibilization effect due to the presence of styrenic moieties, but also because of its elastomeric nature. Results are also less dispersed.

Figure 10 represents unnotched counterparts of a previous figure. Here, all candidates, except for PP-g-SAN again, improve ABS-PP properties back to virgin reprocessed ABS level and even above. Again, 1 phr PP-g-SAN is rather equivalent of the non-compatibilized batch and then PP-g-SAN extend properties degradation, going as low as 25 kJ/m^2^, the equivalent of 6 wt% PP contaminated ABS (see Figure 3).

On other batches, the concentration effect seems nonexistent, maybe except for ABS-g-MA. Here, TPE-g-MA seems to be slightly above the reference, indicating an impact reinforcement effect. The effect is more evident and more pronounced with SEBS, especially as almost half of the samples from all batches (circled in pink) led to partial break (noted “P” in ISO 179 standard). A partial break of the specimen means that a solid and stiff ligament, here at least 1 mm thick, links the two fragments whereas a hinge is flexible, easily broken and very thin, here about 100–200 µm, “skin” thickness. As start velocity was at 2.9 m/s and total mass of the impactor at 3.14 kg, impact energy is theoretically 13.2 J. However, maximal measured energies here are about 7 J, which is significantly inferior. This means that specimens were ejected before the impactor could communicate the energy needed to break it.

Force–displacement curves of unnotched Charpy impact on references and SEBS and PP-g-SAN additive batches are presented in Figure 11. The equivalent for all batches and both impact test types is given in *Appendix A* (*Appendix A*). SEBS lengthens the plateau further than the virgin reference, explaining the higher break energies of Figure 10. A slight force decrease in the plateau is even observed. About half of presented curves are linked to partial break but do not present a different appearance. These curves are just generally longer. Separate curves for each batch and a picture of a partially broken specimen are available in *Appendix A* (*Appendix A*). Interestingly, some totally broken specimens have curves very close to some partially broken ones. Thus, their “break” energies are close as well. Other compatibilizers (*Appendix A*) led to curves reminiscent of virgin ABS, except for ABS-g-MA, slightly less performant. Finally, the PP-g-SAN (Figure 11) effect on unnotched impact curves is very similar to the effect of PP, explaining obtained energies.

#### 3.2.2. Morphology

SEM pictures of cryofractured dogbones of references and 3 phr additive systems are displayed in Figure 12 (5000× magnifications here, 25,000× available in *Appendix A*). As explained before, image analysis is challenged by the rough fracture surface. However, it seems that PP nodules are smaller with ABS-g-MA and TPE-g-MA than on uncompatibilized ABS/PP. Simple measurements gave diameters around 1–2 µm on ABS/PP. With ABS-g-MA and TPE-g-MA, sizes were found around 0.5–1.5 µm. With PP-g-MA and SEBS, nodules were harder to see as they are smaller. They were measured at, respectively, 0.5–1.0 µm and 0.2–0.6 µm. Size diminution could be imputable to interfacial tension decrease as these additives play a tensioactive role as their chemical nature is dual. Considering chemical affinities, it is rather logical that PP-g-MA and SEBS display the best results. Finally, with PP-g-SAN, PP nodules seem surprisingly unaffected.

Figure 13 displays a closer magnification of 1 and 3 phr PP-g-SAN additive system. The SEM pictures were taken within 150 µm from the surface, thus within the skin. Elongated phases were found exclusively in all analyzed samples containing PP-g-SAN. This phase is thus probably PP-g-SAN which was segregated from ABS. These elongated phases were found parallel to the surface and close to it. They were typically found less than 150 µm from it. Interestingly enough on this picture, this new phase seems to have contained what could have been nodules. Since it is mainly found near the surface of specimens, it is more detrimental to unnotched tests since in standard notched tests the notch is 2 mm deep. In this way, one could assume that it could not play a weakening role.

It is surprising to see the PP-g-SAN demixing in this way. Since PP nodules do not appear to be affected by the presence of PP-g-SAN, it could be presumed that most of this additive is found in these segregated phases. However, PP-g-SAN should be compatible to ABS considering its chemistry. Thus, the mixing failure is probably rooted in two different viscosities because of very poor solubility of the additive system in the ABS phase. Modulations of process parameters are performed in the next part to try improving this polymer incorporation.

#### 3.2.3. Tensile Properties

From impact-resistance values and morphology refinement, it was found that PP-g-MA (homopolymer and copolymer undifferentiated) and SEBS are the most promising materials. The second one is more performant but the first one is more common and cheaper. However, tensile properties could be negatively modified, especially by SEBS. They were then assessed for these two additives at different applied concentrations (results in Figure 14 and force–displacement curves in *Appendix A*). Most batches led to results very similar to references which were already close to each other. Notably, properties do not visibly change with the concentration.

#### 3.2.4. Conclusions on Compatibilization Trials

Except for PP-g-SAN, all compatibilizers proved to have an interesting effect, partially restoring impact resistance through morphological refinement, probably due to interfacial tension decrease and coalescence prevention. PP-g-SAN was found to be segregated to the skin, devoid of PP. Additionally, PP nodules seemed unaffected. Finally, it rendered the material even more brittle than with PP alone. All polymers grafted with maleic anhydride had similar effects. In particular, PPH-g-MA and PPC-g-MA behaved very similarly. SEBS had the most important effect of morphological refinement and its quality of impact modifier even led to a partial break during unnotched Charpy impact tests. For the sake of simplicity, PPH-g-MA and SEBS were kept as interesting candidates because of their widespread industrial use, compared to ABS-g-MA for instance. Tensile properties were also shown to be almost unmodified by the compatibilizer presence.

### 3.3. DOE-RSM: Influence of Process Parameters and Interactions with Compatibilizers toward Impact Properties

The following results relate a Design of Experiment (DOE) with Response Surface Methodology (RSM) applied to the previous system to evaluate possible influences and interactions with selected compatibilizers from process parameters on impact properties as they are the most sensible properties of this system. As SEBS and PP-g-MA gave the most promising results and PP-g-SAN was found to be poorly incorporated into the matrix, they were selected for this DOE. The purposes were to observe if properties could be further improved for PP-g-MA and SEBS, and if PP-g-SAN could be better incorporated. As easily tunable parameters, extruding temperature and screws rotation speed were chosen as variables. For the sake of simplicity, extrusion temperature was kept homogenous along the screw, except for the feeding zone, kept at 80 °C. Extrusion temperatures and screw speed limits were, respectively, 200 to 240 °C and 200 to 300 rpm. Following previous results, which showed that PP-g-SAN and PP-g-MA were surely added in excess and that the SEBS effect was only marginally dependent on its loading rate, 0.5 to 1.5 phr were chosen for the first ones and 1.0 to 3.0 phr for the last one as DOE limits. Experimental results were used to build the models as averages and standard deviations. Obtained response surfaces are presented below in 2D projections of isoresponse lines. In *Appendix A*, response surfaces are given in 3D projections, one figure by impact test type (notched or unnotched), enabling better visualization but forbidding objective interpretation.

The Nemrodw^®^ analysis of the RSM study showed that the polynomial model with interaction terms is significant (with a significant p-value of 1.69 for the six responses). The coefficient of multiregression equation was calculated at 0.966 (close to 1) with an average standard deviation between 0.07 and 0.22 kJ/m^2^ for notched impact and between 1.5 and 4.9 kJ/m^2^ for unnotched impact. Averages were, respectively, between 11.47 and 13.69 kJ/m^2^ for notched, and between 49.5 and 100.3 kJ/m^2^ for unnotched. The data were then fitted to a first order polynomial equation with interaction between the terms Xi and Xi.Xj, which validated the model overall response surface of resilience 1eA and 1eU. Finally, this allows us to intrapolate (within the model domain, whereas interpolate is strictly between experimental points) a real simulation of the parameters of extrusion temperature (X1) and screw speed (X2) on the whole of the DOE for the three ratios of PP-g-MA, SEBS and PP-g-SAN compatibilizers. The most important interactions were globally observed between temperature and screw speed and in a lesser measure with compatibilizer rate, even insignificant in some cases. Therefore, it was chosen to represent response surfaces at fixed concentrations, with temperature and screw speed as variables.

#### 3.3.1. Uncompatibilized System

Parameter modulation already leads to impact resistances modifications in uncompatibilized systems (Figure 15). For notched impact, break energies are at 12.4–12.6 kJ/m^2^ for lowest screw speed on all ranges, and highest temperature for all ranges as well. Then, energy increases progressively by following the diagonal line, with decreasing temperature and increasing screw speed, going up to 13.4 kJ/m^2^ in the top left-handed corner. High screw speed can lead to better dispersion as shear rate increase leads to capillary number increase. Lower temperature can prevent material degradation. A lower temperature can also affect viscosity ratios of both phases but a complete rheological assessment is necessary to support this statement. Morphological characterization via SEM (*Appendix A*) was however not conclusive on a potential refinement.

For unnotched Charpy impact, the overall behavior is quite different as evolutions are not homogeneous on the whole ranges, going from 72 kJ/m^2^ from the top right corner to 83 near the center to 90 kJ/m^2^ on a large area at the top left corner. On the right-handed half, corresponding to 220–240 °C, energies increase by diminishing screw speed and temperature simultaneously whereas on the other half, 200–220 °C, energies are increased by diminishing temperature while increasing screw speed. At the extreme bottom of the range (very low screw speed), increases change directions and go toward the corners. These values are close to the domain limits and the validity of the model is thus questionable whereas the top left corner displays a large area with homogenous response.

Unnotched Charpy impact adds skin effects and crack initiation to what is observed in notched impact. These supplementary behaviors are surely due to different skin effects. Indeed, a skin devoid of PP was systematically observed. Its thickness is probably a consequence of the blend dispersion conditions, especially as ABS-PP system viscosity is probably the result of PP domain sizes. Since stress is concentrated at the bottom of the specimen during the impact, as in a flexion test, a thicker skin could enable better properties. To the contrary, poor cohesion between the skin and the core, as seen on Figure 8, could have a negative influence on properties.

Nonetheless, optimization of both properties is achieved by going to the top left corner, with the highest screw speed and lowest temperature. Thus, 3D surfaces (*Appendix A*) display very similar overall shapes for uncompatibilized system for both types of impact.

#### 3.3.2. PP-g-SAN

Figure 16 shows the same response surfaces as those previously seen on ABS/PP/PP-g-SAN systems at 0.5, 1.0 and 1.5 phr. All energies are globally still below the uncompatibilized system, ranging from 8.5 to 12.5 kJ/m^2^ for notched impact instead of 12.6–13.4. This shows a global failure at incorporation attempts. A common break energy maximum about 12.1–12.5 kJ/m^2^ for all rates can be seen for maximal temperature (240 °C) and minimal screw speed (200 rpm). At 0.5 phr, energy is almost independent from temperature and overall is rather constant as only 0.5 kJ/m^2^ evolutions are seen. With higher loading rates, lower temperatures and higher screw speeds lead to stronger decreases down to 8.5 kJ/m^2^ for the highest speed and concentration and lowest temperature. Additionally, the more PP-g-SAN is added to the system, the worse the issue is, probably linked to more segregation occurring in the system.

Similar behavior is seen on the unnotched test, with stronger energy variations on the studied ranges, from 50 to 80 for 0.5 phr. At 1 phr, the top left corner is at 0 kJ/m^2^ whereas the maximum rises from 80 to 90 at the bottom right corner. Null and negative values have no physical signification, and this proves that such a model cannot be applied to such a complex system with such material heterogeneity and so few experimental data, especially when approaching model boundaries. Additionally, even at 0.5 phr, 240 °C and 200 rpm, which is supposed to be optimal here, elongated phases were found near specimens’ surface (*Appendix A*). At 1.5 phr, invalid values occupy roughly a third of the range. A maximum is found at 100 kJ/m^2^ at the lowest speed and highest temperature.

Eventually, even experimental results (40–70 kJ/m^2^) are not satisfactory and do not suggest significant ameliorations above the uncompatibilized system through process alone. Especially, temperature should be increased, which could lead to the material degradation. PP-g-SAN itself should be modified, in molecular weight and/or PP/SAN ratio, to tune its rheological behavior. Otherwise, work on screw profiles could enable better dispersion.

#### 3.3.3. PP-g-MA

PP-g-MA additive systems display rather different behavior (Figure 17) and trends greatly change with incorporation rate, especially beyond 0.5 phr, maybe indicating it is then in excess. Interestingly, notched impact energy ranges are the same for all three rates, roughly 12.2–13.0 kJ/m^2^ despite behavior differences. At 0.5 phr, energy mainly increases following lower temperatures. Similar to what was found with unnotched impact on uncompatibilized ABS/PP, screw speed lowers at the 220–240 °C range but increases at the 200–220 °C range, with a maximum in the top left corner. At 1.0 phr, parameter influences are permuted and screw speed becomes more important and should be lowered. By doing so, the maximum is relocated at the bottom right corner. It can be seen that PP-g-MA notched impact properties’ surface responses are globally lower than those of the uncompatibilized system, whereas experimental data (given in *Appendix A*) seem to indicate the opposite. These contradictory trends are to be linked to the relatively important dispersions of PP-g-MA systems. The Nemrodw^®^ software takes the standard deviation as a value modulation range to reach the best correlation. Thus, here, it was to the disadvantage of PP-g-MA.

For unnotched impact, behaviors at different rates are more homogenous and the general trend is toward low speed and low temperature with rather equal importance of both parameters. However, for 0.5 phr, isoresponse curves go along the axis, respectively, for lowest temperatures and lowest screw speed. As for the uncompatibilized system, this corresponds to the model limits and its reliability is thus questionable. Here, increasing compatibilizer rates improves resistance, whereas results from the previous part shows that 3 phr represent an excess. Optimal rate of PPH-g-MA for 4 wt% PP is therefore probably between 1.5 and 2.0 phr. An optimal compromise between notched and unnotched properties is located around 210 °C and 200 rpm where both values are at the maximum at 1.5 phr, reaching, respectively, 13 kJ/m^2^ and 116 kJ/m^2^. Temperature and screw speed are to be kept at minimum here probably to prevent ABS degradation. PP dispersion could be good enough because of compatibilization and higher shearing would not affect it further as interfacial tension is significantly low enough. Otherwise, Plochocki et al. [48] suggested that shear rate could have an important effect on collision probability, and thus coalescence, in uncompatibilized systems. However, it could be expected from selected compatibilizers that they prevent coalescence through steric stabilization by being situated at the interface (Sundararaj and Macosko, 1995).

#### 3.3.4. SEBS

SEBS results (Figure 18) show behaviors closer to what the uncompatibilized system had, especially for notched impact. Top-left corner is optimal for notched properties and this is amplified by the loading rate. Additionally, values on the whole range are improved due to the high compatibilizer content. Consequently, values go from 13.0 to 14.2 kJ/m^2^ for 1 phr, but from 14.0 to 16.0 kJ/m^2^ at 3 phr. Unnotched impact also displays a strong sensibility to temperature, perhaps to be related SEBS thermal sensitivity. Response toward shear is quite different. At 1 phr, the behavior is reminiscent of what was seen on uncompatibilized ABS-PP and 0.5 PP-g-MA. Since uncompatibilized experimental data were used to build these models as well, it could be its repercussion which is seen at lower rates. For higher loading rates, the behavior is very similar to PP-g-MA systems, with higher values on the whole range. Thus, similar behavior explanations could be applied to this system. Values increments are very similar for both compatibilizers, rate by rate. Additionally, a notable effect of compatibilizer loading rate is seen, stronger than with PP-g-MA, with a maximum at 122 kJ/m^2^. This is also very visible on 3D projections in *Appendix A* (*Appendix A*).

Compromise between notched and unnotched is more delicate than with PP-g-MA since notched impact has here a higher sensibility to temperature, probably due to SEBS decomposition which is more sensible to temperature than PP-g-MA. Indeed, isoresponse curves are almost diagonal instead being vertical. The temperature of 200 °C is indubitably optimal but screw speed should be chosen in a between 225 and 275 rpm, depending on which property is preferred.

#### 3.3.5. Conclusions on RSM Results

These experiments showed that impact resistance could be modified by the modulation of extrusion parameters, temperature and screw speed, and compatibilizer type and loading rates. An uncompatibilized ABS + 4 wt% PP system could be optimized with high screw speed and lower temperature. The first one promotes finer dispersion of the minor phase through shear rate and capillary number increases. Temperature minimization can avoid material degradation. PP-g-SAN was not noticeably better dispersed and its properties remained lower than the uncompatibilized system. Rheological assessment could enable us to understand this failure. PP-g-MA and SEBS showed different results regarding notched and unnotched impact resistance. For notched, high screw speed and low temperature are to be preferred whereas low screw speed and low temperature are better for unnotched. This highlights difference in crack initiation and potentially skin effect. However, lower shear and temperature can prevent ABS degradation. Overall SEBS is confirmed as the most performant candidate, imputable to its dispersing role, but also its impact modifier role. In-depth rheological assessment of this system could be interesting to better highlight the occurring phenomena. For the sake of the economy of the experiment, all systems were incorporated into the same DOE. In the future, quadratic models (individual factors taken at 2nd degree) could be applied to these systems for better representativity of physico-chemical reality, especially for PP-g-SAN because of its lack of significance.

## 4. Conclusions

PP was incorporated in ABS through twin-screw extrusion at different rates: 2, 4, 6 and 8 wt%, in order to evaluate and manage the effect of residual impurities in WEEE plastics. It induced important losses of impact resistance, from both notched and unnotched Charpy impact. Tensile properties were unmodified at 2 wt% but they decrease above, with a noticeable weakening above 6 wt%. Morphology assessment showed a nodular structure. Additionally, important heterogeneities were observed at the millimetric scale above 6 wt% of PP.

Several potential compatibilizers were then tested on ABS contaminated with 4 wt% PP since properties began to be severely affected from this fraction. PP-g-MA and SEBS were found to be the most interesting, the first one which is relatively usual led to interesting properties, the second one enabled obtaining the best results of the study. Both entailed an important morphological refinement effect as proven by SEM observations. Tensile properties were unaffected by both compatibilizers. PPC-g-MA, ABS-g-MA and TPE-g-MA had similar effect than PPH-g-MA but were less interesting as they are less usual and commercially available. PP-g-SAN was found to be segregated. It did not modify the morphology and even led to a weakening of the material.

Design Of Experiment (DOE) were applied on ABS + 4wt% PP system with compatibilizer (none, PP-g-MA, SEBS, PP-g-SAN), compatibilizer rate, extrusion temperature (200–240 °C) and screw speed (200–300 rpm) as variables and impact properties (both notched and unnotched) as responses. The experimental data were fitted to a first-order and second-order interactions polynomial equation, which validated the model overall response surface of notched and unnotched impact resistances. Response Surface Methodology (RSM) revealed that properties of an uncompatibilized system could be enhanced, but still far from virgin ABS, with stronger screw speed and lower temperature. PP-g-MA remained segregated, resulting in poor results. Blends with PP-g-MA and SEBS were found to be optimized, especially for unnotched impact. Lower temperature and milder screw speed are generally to be advised probably to prevent ABS degradation. Since PP could be already adequately dispersed, stronger shearing could be unnecessary.

This study was based on virgin samples from the same grade. In a real recycling context, different grades with various formulations and different degradation degrees are mixed together, leading to specific viscosities and mechanical properties. As highlighted with ABS contaminated with 8 wt% PP, multi-scale segregation can occur, probably inducing mechanical weakness. Several research and development pathways can follow these works. Additionally, different types of SEBS exist, with different S/EB ratios, rheological behavior; they also impact modification ability. Other process parameters could be considered in DOE as feeding rates and models could be refined through more experimental data. Finally, it could be interesting to better assess created morphologies with other sample preparation methods and then evaluate their durability through annealing tests.

## Figures and Tables

**Figure 1 polymers-13-01439-f001:**
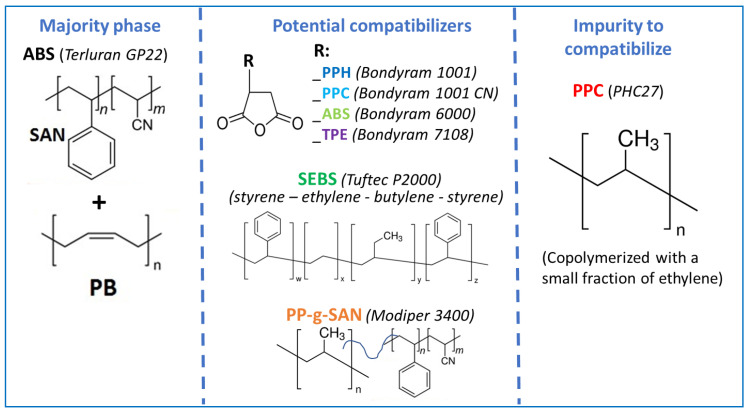
Materials chemistry.

**Figure 2 polymers-13-01439-f002:**
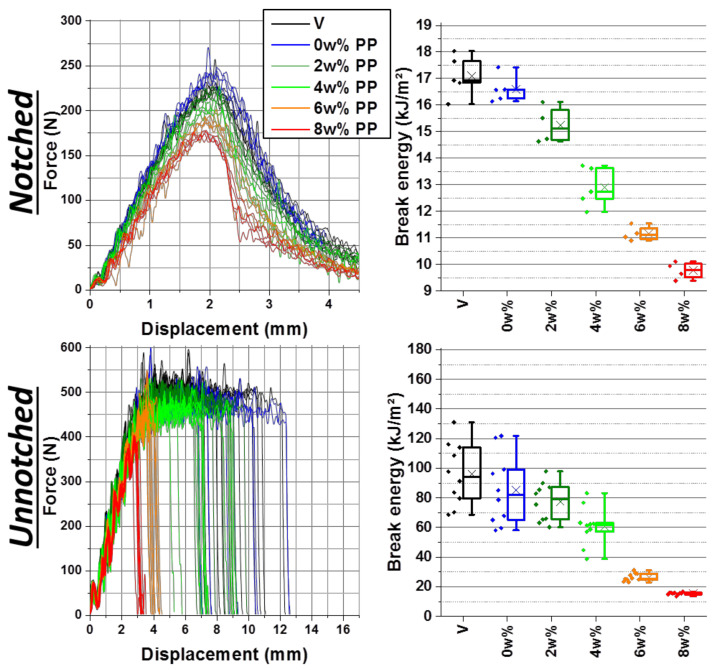
Force–displacement curves (**left**) and break energies boxplots (**right**) of notched (**top**) and unnotched (**bottom**) Charpy impact tests on ABS contaminated at increasing rates of PP —“V” for directly injected ABS, “0 wt%” went through extrusion.

**Figure 3 polymers-13-01439-f003:**
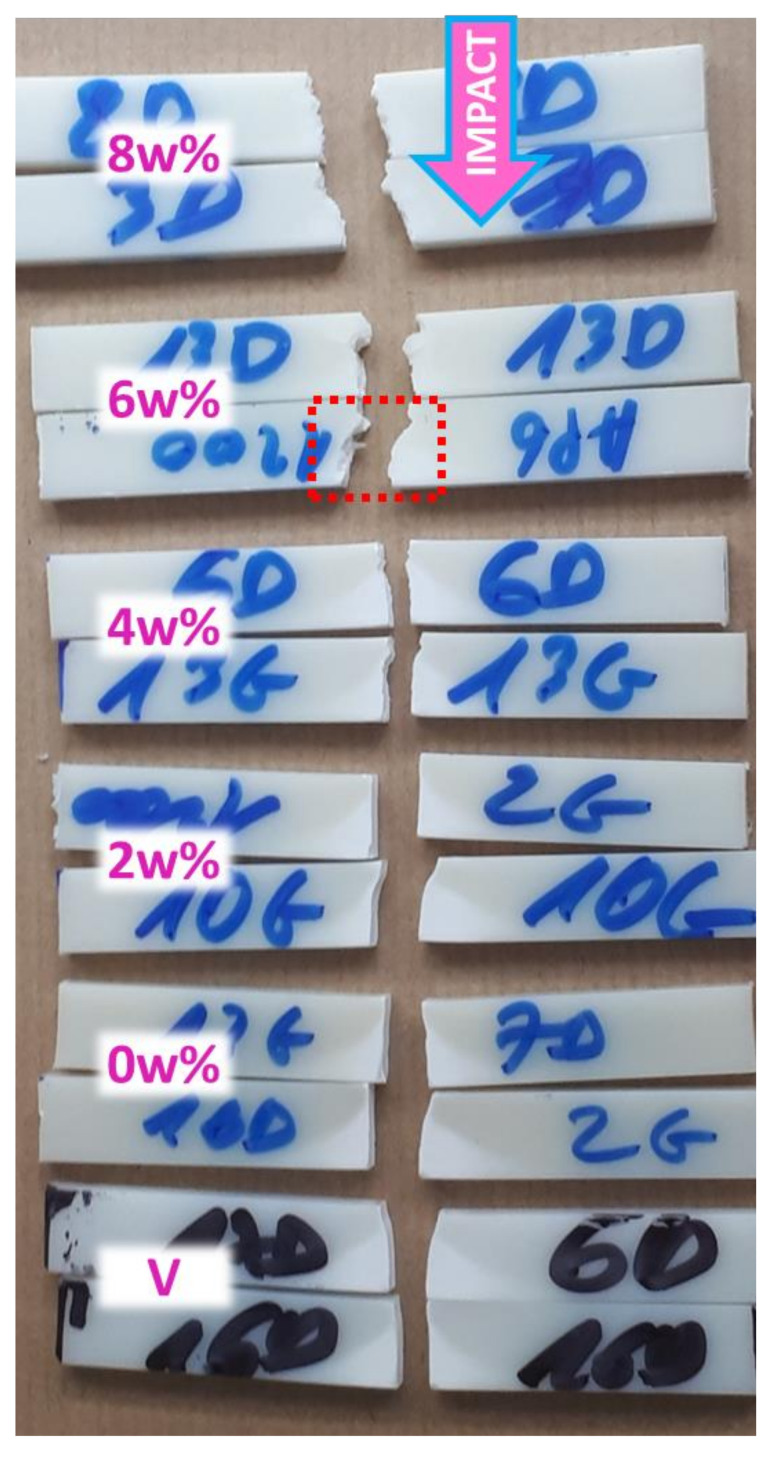
Picture of unnotched Charpy impact broken specimens of ABS gradually contaminated with PP—arrow shows direction of impact; rectangle points out dephasing at macro-scale.

**Figure 4 polymers-13-01439-f004:**
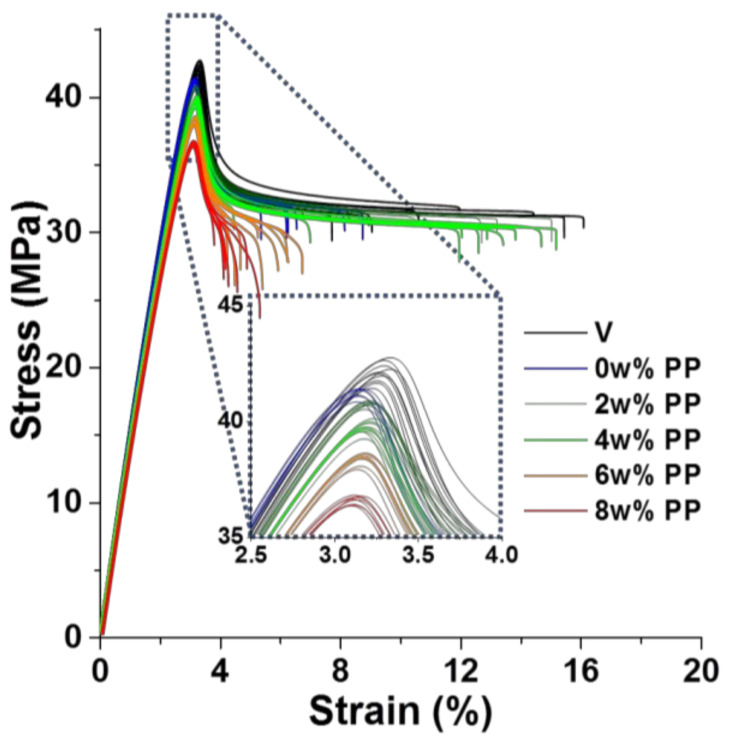
Stress–strain curves of tensile tests on ABS contaminated at increasing rates of PP—magnification on stress peak “V” for directly injected ABS, “0 wt%” went through extrusion.

**Figure 5 polymers-13-01439-f005:**
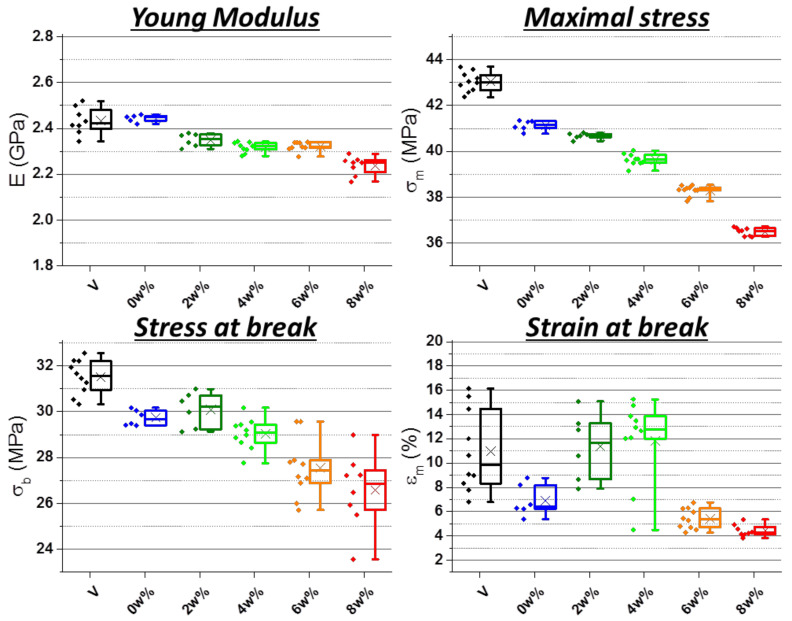
Young Modulus, maximal stress, stress and strain at break from tensile tests on ABS contaminated at increasing rates of PP—“V” for directly injected ABS, “0 wt%” went through extrusion.

**Figure 6 polymers-13-01439-f006:**
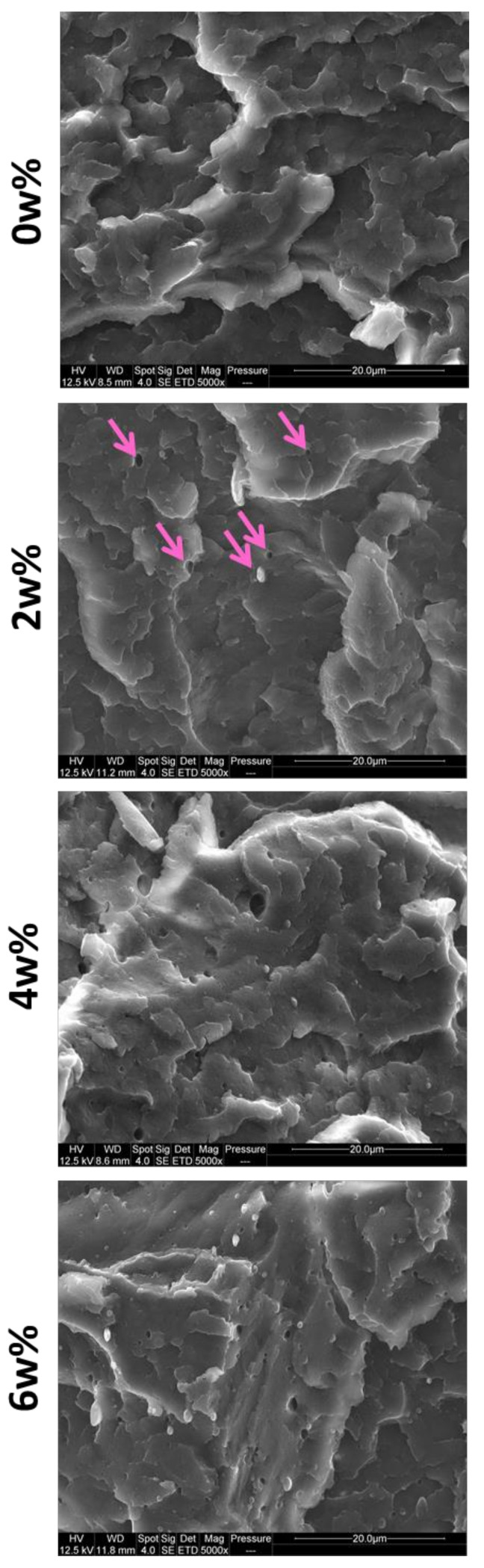
SEM pictures of cryofractured ABS dogbones contaminated at 0, 2, 4 and 6 wt% of PP—5000× magnification.

**Figure 7 polymers-13-01439-f007:**
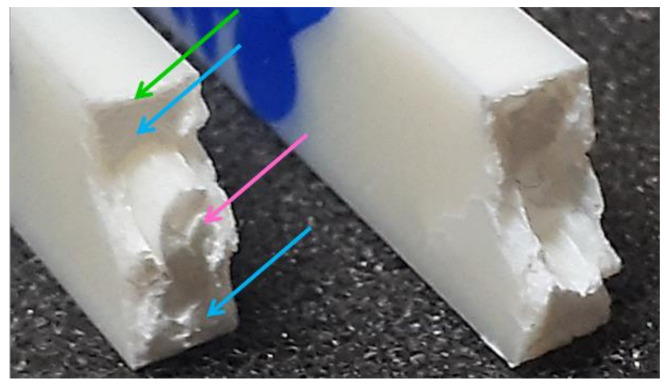
Picture of a Charpy post-mortem dogbone of ABS + 8 wt% PP—arrows indicate different areas.

**Figure 8 polymers-13-01439-f008:**
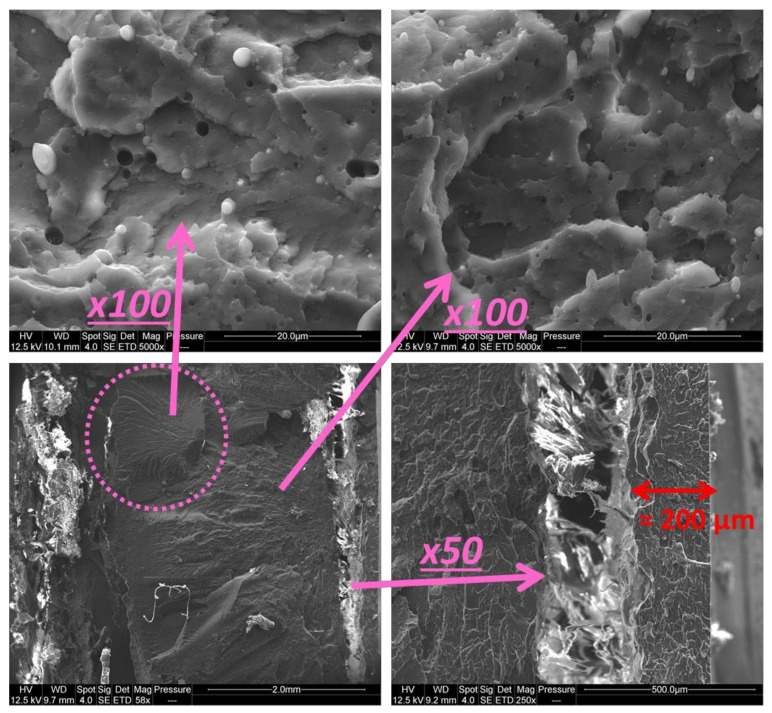
SEM pictures of cryofractured 8 wt% PP contaminated ABS dogbone—magnified views of different parts.

**Figure 9 polymers-13-01439-f009:**
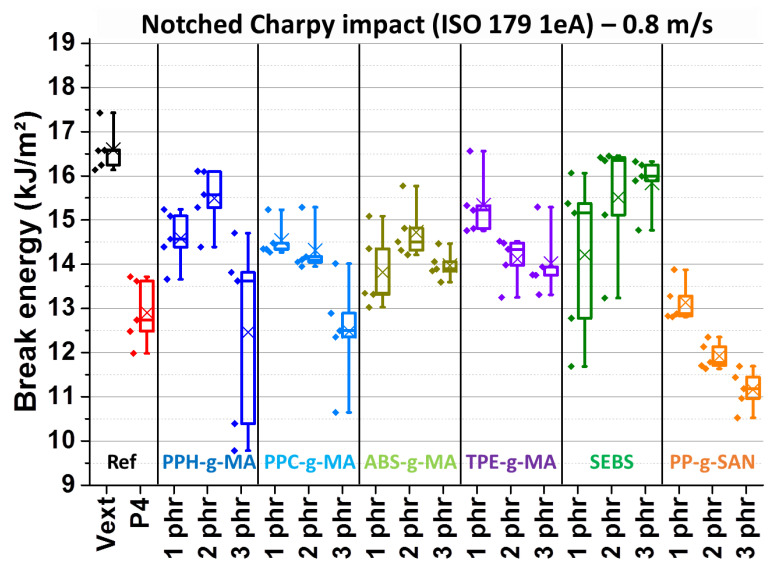
Notched Charpy impact break energies—virgin extruded ABS batch “Vext”, 4 wt% PP contaminated batch “P4”and separately added batches with 1, 2 and 3 phr of PPH-g-MA, PPC-g-MA, ABS-g-MA, TPE-g-MA, SEBS and PP-g-SAN.

**Figure 10 polymers-13-01439-f010:**
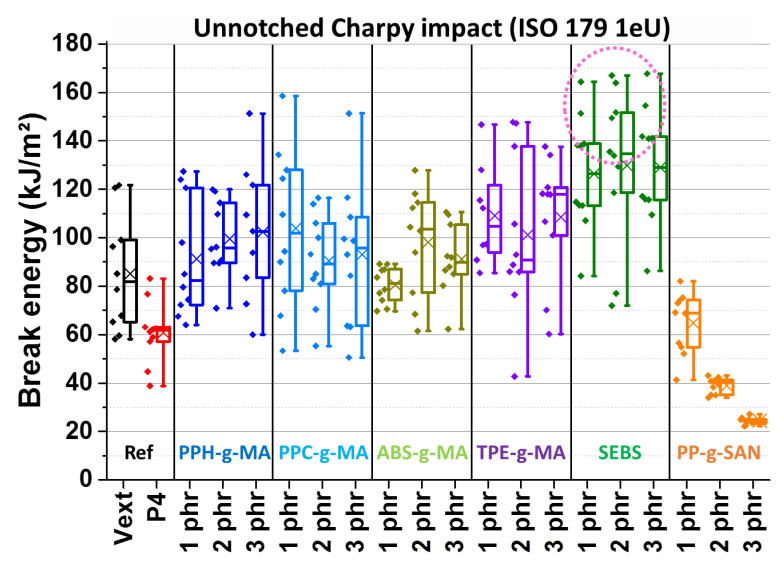
Unnotched Charpy impact break energies—same batches and denominations as previous figure—circle for partial breaks, respectively, 4, 5 and 4 specimens for 1, 2 and 3 phr of SEBS.

**Figure 11 polymers-13-01439-f011:**
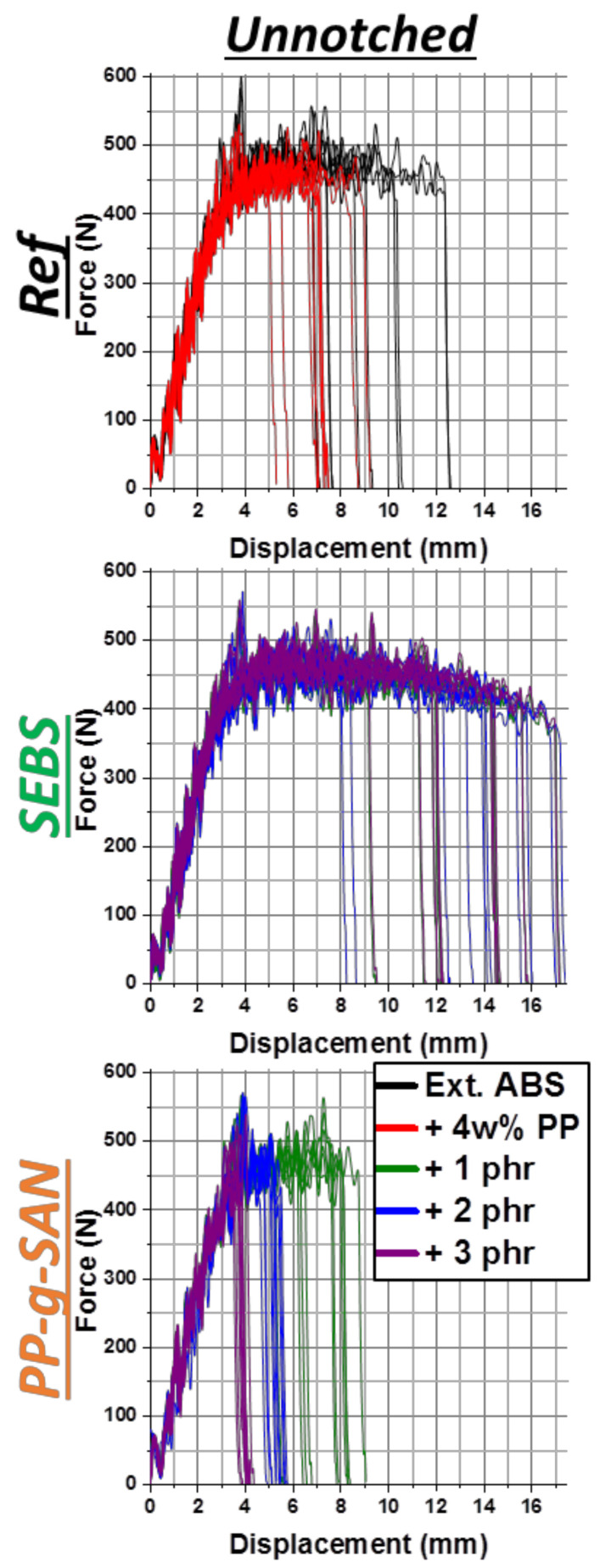
Force–displacement curves of unnotched Charpy impact of ABS—virgin and 4 wt% PP contaminated (1st line), added with 1, 2 and 3 phr of SEBS (2nd line) or PP-g-SAN (3rd line)—color associated with additive rate.

**Figure 12 polymers-13-01439-f012:**
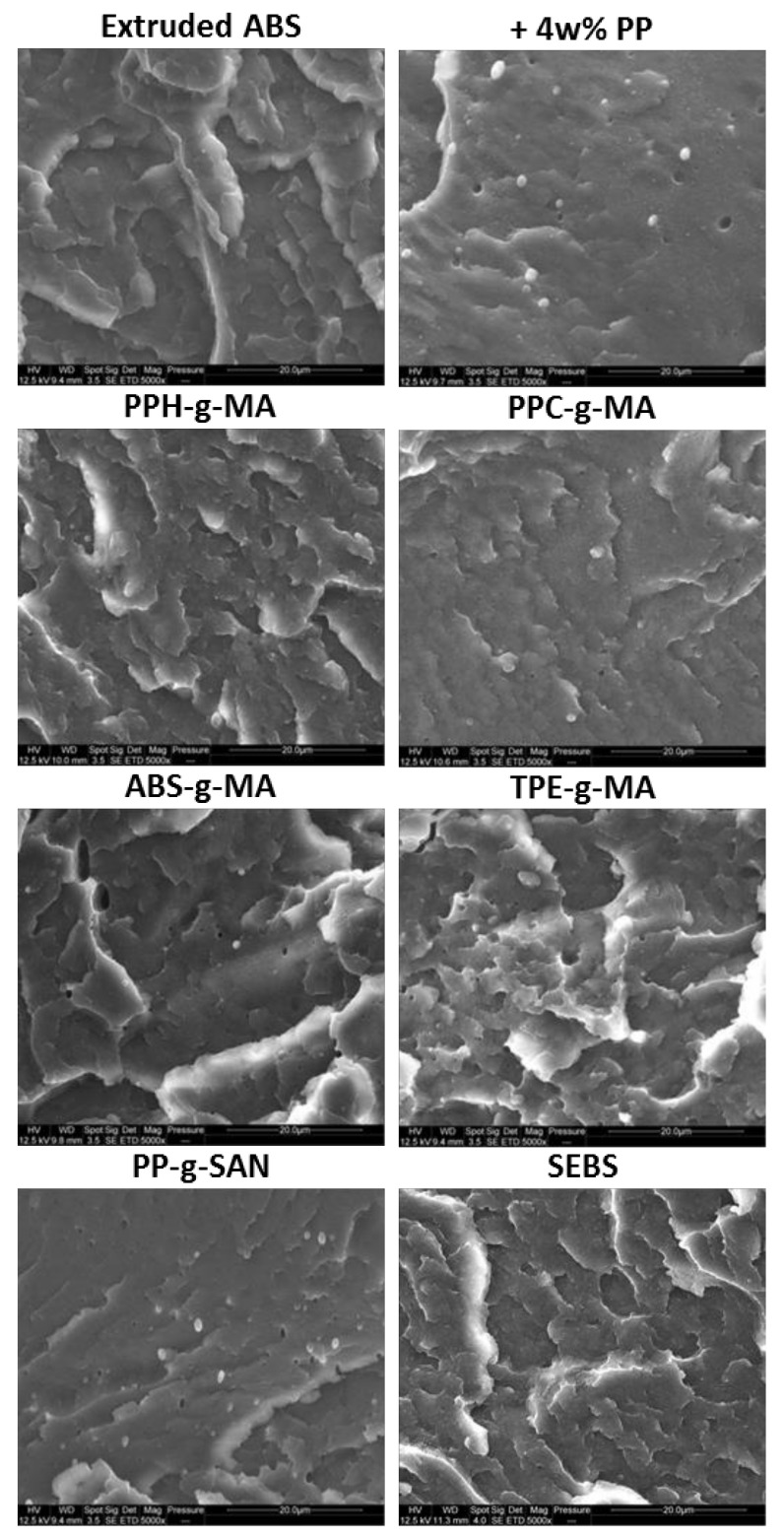
SEM pictures of cryofractuted dogbones—effect of compatibilizers at 3 phr on a 4 wt% PP contamination—5000 × magnification.

**Figure 13 polymers-13-01439-f013:**
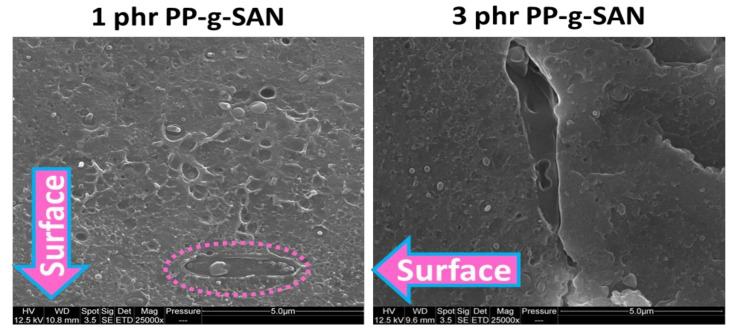
SEM pictures of cryofractured dogbones of ABS contaminated with 4 wt% PP and added with PP-g-SAN at 1 and 3 phr—25,000× magnifications—pictures taken within 100 µm from the surface—arrows indicates surface direction—supposedly PP-g-SAN phase.

**Figure 14 polymers-13-01439-f014:**
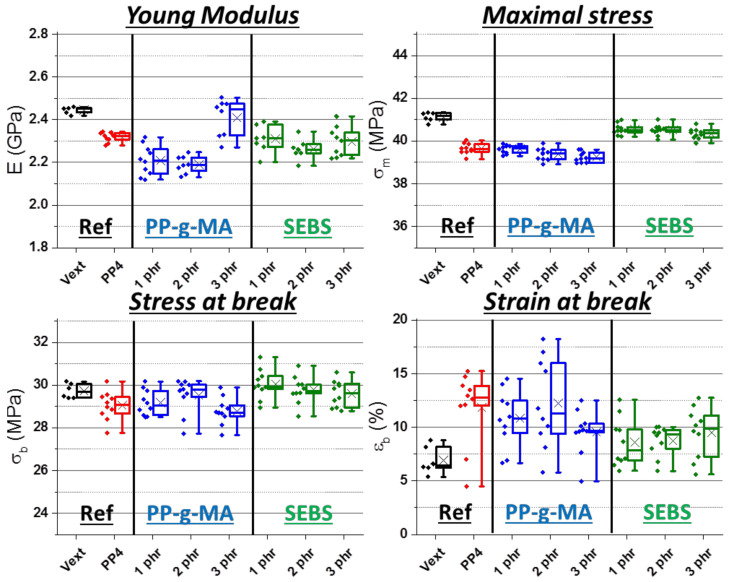
Young’s modulus, maximal stress, stress and strain at break from tensile tests on ABS contaminated with 4 wt% of PP—“Vext” for virgin and extruded ABS, other batches with 4 wt% PP—phr rates correspond to PP-g-MA and SEBS.

**Figure 15 polymers-13-01439-f015:**
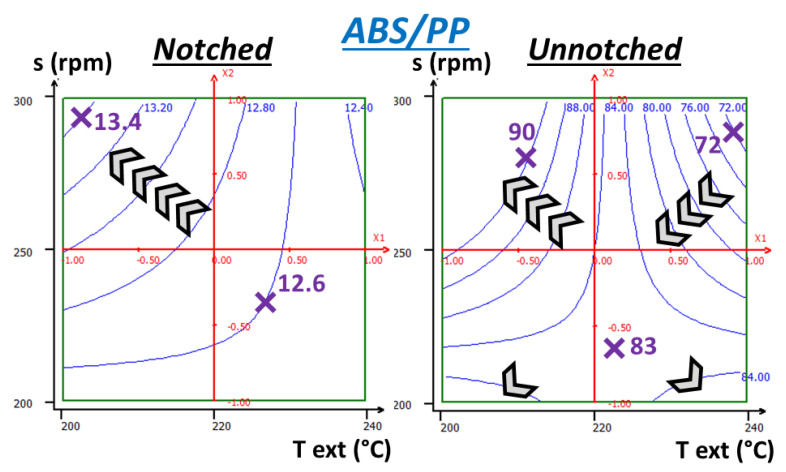
RSM applied to notched and unnotched Charpy break energies (kJ/m^2^) of ABS + 4 wt% PP—extruder temperature (200–240 °C) and screw speed (200–300 rpm) as variables—chevrons indicate increasing energy direction.

**Figure 16 polymers-13-01439-f016:**
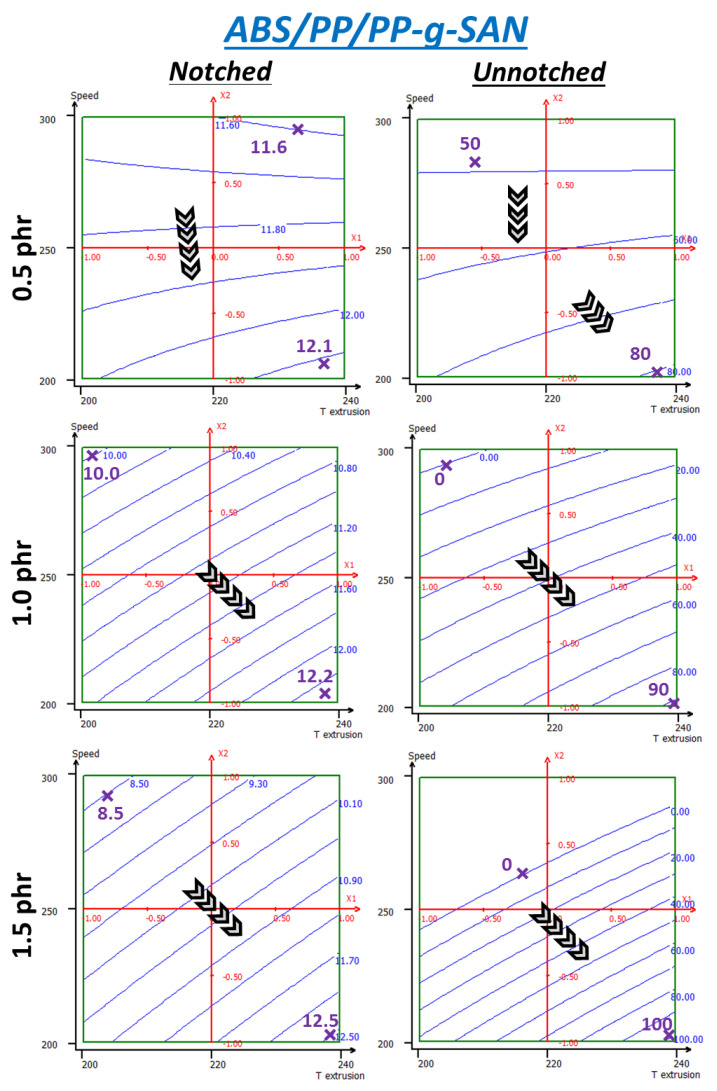
RSM applied to unnotched Charpy break energies (kJ/m^2^) of ABS + 4wt% PP + PP-g-SAN—same variables as those used previously.

**Figure 17 polymers-13-01439-f017:**
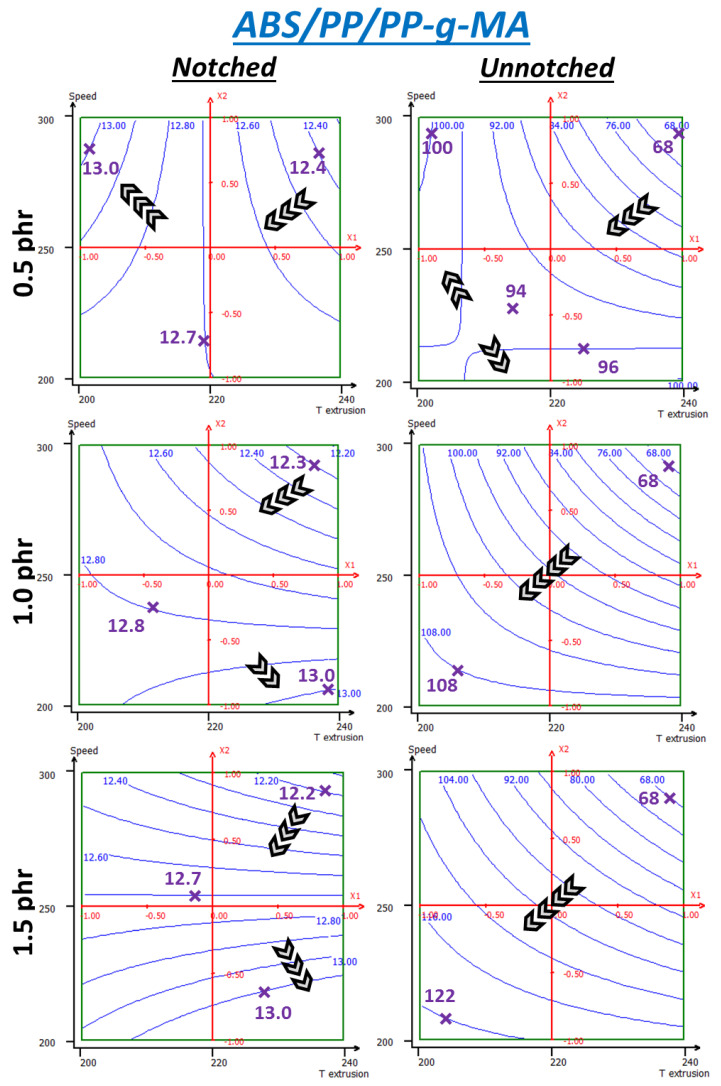
RSM applied to notched and unnotched Charpy break energies (kJ/m2) of ABS + 4wt% PP + PP-g-MA—same variables as those used previously.

**Figure 18 polymers-13-01439-f018:**
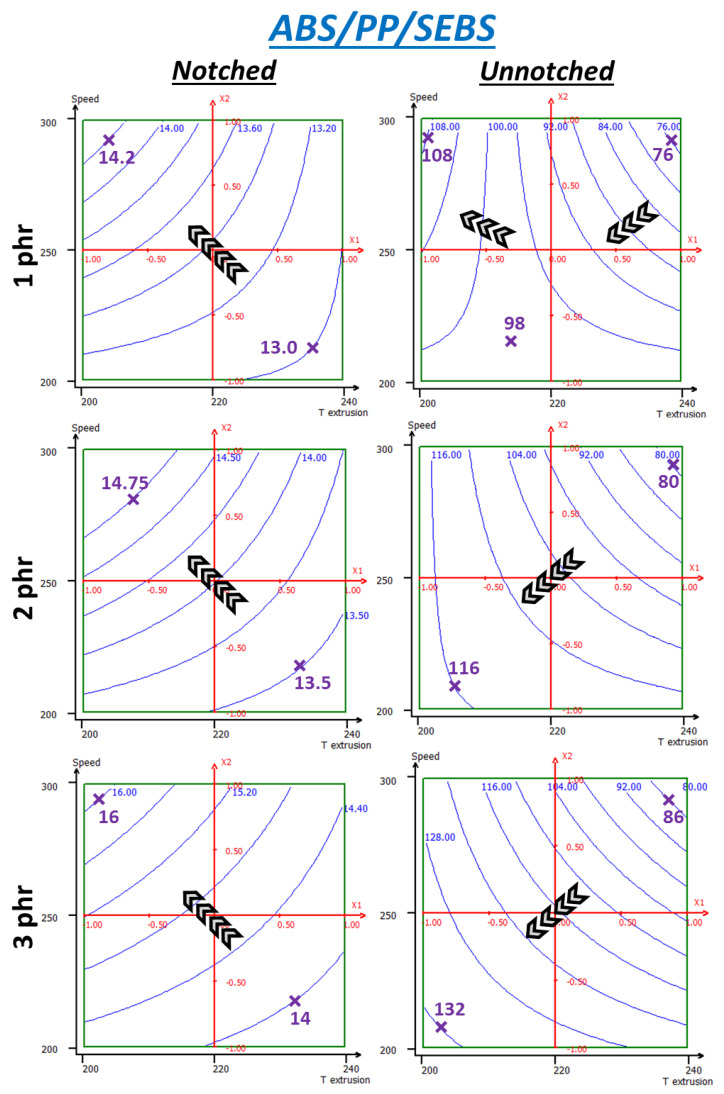
RSM applied to notched and unnotched Charpy break energies (kJ/m^2^) of ABS + 4 wt% PP + SEBS—same variables as those used previously.

## Data Availability

Not applicable.

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
