# Peer review of "Degradation of Styrenic Plastics during Recycling: Accommodation of PP within ABS after WEEE Plastics Imperfect Sorting"

_polymers, 2021, doi:10.3390/polym13091439_

Round 1
Reviewer 1 Report
The topic of the paper - studying the properties of recycled plastics from waste electrical and electronic equipment, taking into account the presence of residual impurities - is really interesting, modern, and important. However, this work provides an example of how not to write an article. First of all, the literature references appear in the form of "Error! Reference source not found". Page numbering is interrupted and starts again. This is not critical, but it gets in the way. The article does not offer any new theoretical ideas and does not discuss old ones. Interpolation of the experimental results by a polynomial is proposed as the only theoretical "model", but it is not said how its coefficients are found and why it is needed. This formula should be shifted to where it is used.
So - the typical reader of this article is a "consumer" engineer. He has the right to expect specific recommendations for specific substances or their classes. Instead, the entire Conclusion is written in the spirit: "PP-g-MA and SEBS 377 were found the most interesting, the first one which is relatively usual led to interesting properties, the second one enabled to obtain the best results of the study." "Interesting" can not help the reader in any way! The section "Conclusions on RSM results" contains a literature review, which it would be desirable to shift to the Introduction.
In general, for this paper to be useful to both its readers and authors, it must be substantially rewritten - structured, the main results and their discussion highlighted, and practical recommendations for their practical use should be given.
Author Response
|
Reviewer #1’s comments |
Authors’ answers |
|
1) First of all, the literature references appear in the form of "Error! Reference source not found". Page numbering is interrupted and starts again. |
Lines and page numbering was corrected. All automated links issues were also corrected. |
|
2) Interpolation of the experimental results by a polynomial is proposed as the only theoretical "model", but it is not said how its coefficients are found and why it is needed. This formula should be shifted to where it is used. |
A further explanation was given in paragraph 2 and a displacement of the model-specific equation was made as requested. |
|
3) So - the typical reader of this article is a "consumer" engineer. He has the right to expect specific recommendations for specific substances or their classes. |
Text was made clearer. |
|
4) The section "Conclusions on RSM results" contains a literature review, which it would be desirable to shift to the Introduction. |
We assume the comment concerns the “3.3.3. PP-g-MA” section as no literature references are found elsewhere in the RSM section. The two references were already assessed in the Introduction. Therefore, the two sentences in question were removed. |
|
5) In general, for this paper to be useful to both its readers and authors, it must be substantially rewritten - structured, the main results and their discussion highlighted, and practical recommendations for their practical use should be given. |
Text was largely edited to improve language level and clarity. Consequently, interpretation, main conclusions and practical recommendations are highlighted. |
Reviewer 2 Report
The authors of the article entitled "Degradation of Styrenic Plastics during Recycling: Accommodation of PP within ABS after WEEE Plastics Imperfect Sorting" thoroughly describe the results they recorded. The concept of using the appropriate methodology for planning the experiment to determine the most advantageous selection of a compatibilizer for the non-miscible PP-ABS system is an interesting approach to the problem of processing materials that are problematic from the point of view of recycling. However, it should be remembered that when using already known systems, it is necessary to undertake additional research so that the indirectly observed material effects can be justified not in literature studies but the actual values ​​of the materials processed by the authors. Below I enclose major and minor comments, including the most important one concerning the need to carry out an additional rheological analysis that will allow for an accurate reference to the change in the compatibility of systems processed under different conditions.
- The text should be corrected in terms of the English language, especially the correctness of the translation of terms and phrases used, such as "stronger screw speed" etc. Additionally, the text should be corrected to remove numerous editing errors, such as values ​​combined with units.
- All charts should be corrected to make the fonts uniform and improve their legibility.
- There are numerous errors in the text related to the damaged hyperlinks to the automatic numbering of drawings and references.
- Information on the processing properties of the polymers used should be provided in section 2.1.
- I recommend that the authors limit the number of presented mechanical test results or compile them more concisely. Descriptions of the RSM are significantly too long and contain literature references and general statements that do not provide additional information. What is worse, they are introduced contrary to the journal's formatting guidelines (line 316, 320).
- As the subject of improving the miscibility of compositions with PP is quite extensively discussed in the literature, the research methodology used for its evaluation, i.e. mechanical properties and SEM analysis, is insufficient. It would be necessary to use additional rheological analyses to assess the thermo-rheological properties in terms of changing the miscibility of the systems induced by the coupling agents' presence. Assessment of rheological properties using oscillatory rheology performed at temperatures identical to the applied extrusion conditions will allow for the completion of the discussed results. In its present form, despite the significant amount of data presented and references to other works in the literature, there is no scientific justification for the conclusions drawn.
The article should be significantly revised. I believe that it may be helpful after appropriate corrections from the point of view of compiling significant amounts of data. However, it is necessary to supplement the manuscript with additional information to understand the changes taking place. It does not constitute only a research report and to modify the already presented content to improve its readability.
Author Response
|
Reviewer #2’s comments |
Authors’ answers |
|
1) However, it should be remembered that when using already known systems, it is necessary to undertake additional research so that the indirectly observed material effects can be justified not in literature studies but the actual values ​​of the materials processed by the authors. |
Maybe not clear but it is precisely from the bibliographical references in which the authors discuss their work on experimental tests (actual values) that we based our study. |
|
2) The text should be corrected in terms of the English language, especially the correctness of the translation of terms and phrases used, such as "stronger screw speed" etc. Additionally, the text should be corrected to remove numerous editing errors, such as values combined with units. |
Editing errors were removed. Several turns of phrases and specific terms were changed to try improving the English level. |
|
3) All charts should be corrected to make the fonts uniform and improve their legibility. |
The most of the charts have be corrected to make the fonts uniform. |
|
4) There are numerous errors in the text related to the damaged hyperlinks to the automatic numbering of drawings and references. |
These errors were corrected |
|
5) Information on the processing properties of the polymers used should be provided in section 2.1. |
Information was added in the section 2.1. |
|
6) I recommend that the authors limit the number of presented mechanical test results or compile them more concisely. Descriptions of the RSM are significantly too long and contain literature references and general statements that do not provide additional information. What is worse, they are introduced contrary to the journal's formatting guidelines (line 316, 320). |
Literature references were removed. For mechanical testing, the authors understand that they represent a large amount. However, all presented results serve the discussion and a more concise presentation could affect legibility.
|
|
7) As the subject of improving the miscibility of compositions with PP is quite extensively discussed in the literature, the research methodology used for its evaluation, i.e. mechanical properties and SEM analysis, is insufficient. It would be necessary to use additional rheological analyses to assess the thermo-rheological properties in terms of changing the miscibility of the systems induced by the coupling agents' presence. Assessment of rheological properties using oscillatory rheology performed at temperatures identical to the applied extrusion conditions will allow for the completion of the discussed results. In its present form, despite the significant amount of data presented and references to other works in the literature, there is no scientific justification for the conclusions drawn. |
As said previously by the reviewer, experimental results are already given in large quantities. Accordingly, the present manuscript is already rather long. However, the authors feel that most of the presented results are essential to discuss the subject thoroughly. To be discussed properly, rheological results should be provided in important quantities, which would make the paper too long and more difficult to read as it already is.
In consequence, the authors feel that such an assessment deserve to be presented in a separated work. |